# 360VFI: A Dataset and Benchmark for Omnidirectional Video Frame Interpolation

## Abstract

Head-mounted 360° displays and portable 360° cameras have significantly progressed, providing viewers a realistic and immersive experience. However, many omnidirectional videos have low frame rates that can lead to visual fatigue, and the prevailing plane frame interpolation methodologies are unsuitable for omnidirectional video interpolation because they are designed solely for traditional videos. This paper introduces the benchmark dataset, 360VFI, for Omnidirectional Video Frame Interpolation. We present a practical implementation that introduces a distortion prior from omnidirectional video into the network to modulate distortions. Specifically, we propose a pyramid distortion-sensitive feature extractor that uses the unique characteristics of equirectangular projection (ERP) format as prior information. Moreover, we devise a decoder that uses an affine transformation to further facilitate the synthesis of intermediate frames. 360VFI is the first dataset and benchmark that explores the challenge of Omnidirectional Video Frame Interpolation. Through our benchmark analysis, we present four different distortion condition scenes in the proposed 360VFI dataset to evaluate the challenges triggered by distortion during interpolation. Besides, experimental results demonstrate that Omnidirectional Video Interpolation can be effectively improved by modeling for omnidirectional distortion.

## 1 Introduction

In pursuit of a realistic visual experience, omnidirectional videos (ODVs), also known as 360° videos or panoramic videos, have obtained research interest in the computer vision community and become an essential basis of augmented reality (AR) and virtual reality (VR). To have a continuous and immersive experience, ODVs require an extremely high frame rate. However, because of the high industrial cost of high-precision camera sensors, the frame rates of most ODVs are relatively low.

Traditional Plane Video Frame Interpolation (Plane VFI) techniques have been widely used to address the challenge of low frame rates. Significant progress in these methods has been driven by the development of optical flow networks (Dosovitskiy et al., 2015; Sun et al., 2018; Teed & Deng, 2020; Kong et al., 2021), which allow for accurate frame registration in video sequences by establishing explicit correspondences between frames (Jiang et al., 2018a; Xu et al., 2019; Niklaus & Liu, 2020; Park et al., 2021). These flow-based methods generally follow a three-step process: first, estimating optical flow between two input frames and the target frame; second, warping the input frames or their features using predicted flow fields to achieve spatial alignment; and third, refining these warped frames or features through a synthesis network to generate the final target frame.

Recently, IFRNet (Kong et al., 2022) has advanced this approach by allowing intermediate flows and features to enhance each other, resulting in sharper moving objects and better texture detail. While effective for traditional videos, applying optical flow to omnidirectional video interpolation (ODI) is less efficient. Recent efforts, such as SLOF (Bhandari et al., 2022) and MPF (Li et al., 2022), have adapted optical flow to the omnidirectional domain. SLOF re-projected ERP (equirectangular projection) frames onto a sphere for rotational augmentation, while MPF transformed ERP frames into multiple projection formats and fused them for more precise flow estimation. However, both approaches neglect ERP distortion as prior knowledge,

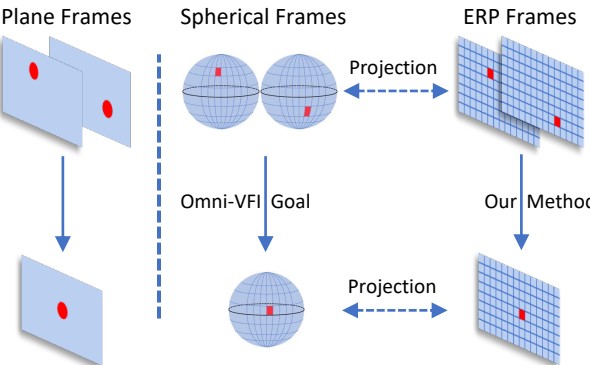

Figure 1: Left: Traditional Video Frame Interpolation. The inputs are two adjacent plane frames, and the output is a target plane frame. Right: Omnidirectional Video Frame Interpolation. The inputs are two adjacent omnidirectional frames with a full field-of-view from an omnidirectional video, and the output is a target omnidirectional frame. Original omnidirectional frames are spherical, and the most common format is the equirectangular projection type (ERP). The two kinds of omnidirectional video formats can be projected from each other, and our proposed method tackles ERP videos.

limiting their efficiency in interpolating omnidirectional frames. Since omnidirectional videos (ODVs) are typically stored and transmitted in the ERP format, introducing significant distortions, traditional frame interpolation methods struggle to handle them effectively. Therefore, directly incorporating ERP distortion as prior knowledge in the interpolation process presents a more effective and efficient solution.

Datasets for plane VFI tasks have been sufficiently researched, including Vimeo90K (Xue et al., 2019), UCF101 (Soomro et al., 2012), and SNU-FILM (Choi et al., 2020). Vimeo90K is the most popular dataset for plane VFI, composed of triplet samples. However, Vimeo90K is not designed for motion extent, and this is very important in omnidirectional video interpolation because of its latitude-dependent distortion. SNU-FILM was proposed for the motion-based interpolation benchmark. Both are focused on Plane VFI. Recently, omnidirectional video datasets have caught the attention of the computer community. Several datasets were proposed for omnidirectional video super-resolution, including ODV360 (Wang et al., 2023), 360VDS, and 360UHD (Baniya et al., 2023). They are collected from either YouTube or by themselves. Therefore, there are no omnidirectional video frame Interpolation (Omni-VFI) datasets that have both sufficient and various samples.

To alleviate these problems, we introduce a dataset for omnidirectional video interpolation called 360VFI dataset. We also propose a benchmark (360VFI), which injects the omnidirectional distortion into the network based on the 360VFI dataset. Specifically, in the 360VFI dataset, our evaluation benchmark has four different settings. We classified the triplets based on vertical motion extent due to the latitude-dependent distortion in the omnidirectional frame. In 360VFI Network, we propose the DistortionGuard module to extract less distorted pyramid features from input frames. Moreover, we propose OmniFTB to refine the distorted ERP target frame gradually. Experimental results demonstrate that omnidirectional video interpolation can be effectively improved by modeling for omnidirectional distortion.

Our contribution can be summarized as follows:

- We present the first omnidirectional video frame interpolation datasets 360VFI. The dataset covers a wide range of content and motion patterns, providing a comprehensive benchmark for evaluating interpolation methods in the omnidirectional domain.

- We propose the first practical implementation of using distortion priors for omnidirectional video frame interpolation, offering a more accurate and efficient approach to handling the inherent challenges of omnidirectional video processing.

- Our experiments demonstrate that 360VFI achieves SOTA performance on the proposed benchmarks by considering omnidirectional priors in both feature extraction and frame generation, especially in the scene of large motion omnidirectional frame interpolation.

## 2 Related Work

### 2.1 Omnidirectional Data Processing

Omnidirectional images and videos have been attracting increasing attention from computer vision and graphics researchers. Due to the special representation of omnidirectional images, dedicated methods were developed for analysis and processing tasks, such as depth estimation (Wang et al., 2020a), salient object detection (Li et al., 2019b; Wu et al., 2022), Visual Localization (Huang et al., 2024), and video stabilization (Tang et al., 2019). With the recent development of immersive technologies, researchers have been working on the specific problems of omnidirectional video-based applications, such as automatic 360° navigation (Kang & Cho, 2019; Hu et al., 2017), omnidirectional video assessment (Li et al., 2019a), and immersive video editing (Nguyen et al., 2017). However, the absence of consistent temporal correspondences poses challenges for various applications, including Omni-VFI. In response to this challenge, our method aims to address the lack of reliable temporal correspondences in omnidirectional video processing. Furthermore, significant research (Esteves et al., 2018; Cohen et al., 2018) has focused on omnidirectional or 360° data, including spherical and ERP data, aimed at learning sphere representations.

Most convolutional networks were initially designed to project images onto flat surfaces like traditional camera sensors. To extend their applicability to spherical images, researchers have proposed various solutions aimed at enabling convolutions on such images to facilitate feature extraction and interpretation by deep networks. A notable method, known as SphereNet (Coors et al., 2018; Yang et al., 2021), adjusts the perceptual field of its convolutional kernels based on the latitude within the ERP domain. However, altering the kernels of 2D optical flow networks to spherical kernels may compromise the effectiveness of pre-trained models. To address this issue, the kernel transform technique (Su & Grauman, 2019) has been explored, allowing convolutional layers to learn how to transform spherical kernels to the pre-trained weights of standard kernels initially trained on perspective images. Nevertheless, the significant number of layers in current optical flow networks poses practical challenges in terms of computation and memory costs associated with learning to transform all layers. Therefore, further research and improvements are necessary to overcome these obstacles and enhance their applicability in the domain of omnidirectional image and video processing.

### 2.2 Video Frame Interpolation

Video frame interpolation aims at generating non-existent frames between two adjacent input frames. In Long et al. (2016), a pioneering learning-based method was proposed that can directly synthesize intermediate frames between two frames. Consequently, frame interpolation methods based on spatially adaptive convolution kernels (Niklaus et al., 2017; Lee et al., 2020; Cheng & Chen, 2021) or pixel phases (Meyer et al., 2015; 2018) were proposed. However, the former leads to large parameters and high complexity, especially when dealing with complex motion. The latter has difficulty handling complex motion due to an inaccurate estimation of phase and amplitude values. After that, many flow-based methods (Jiang et al., 2018b; Xue et al., 2019) use optical flow to guide warping, but the inaccuracy of the predicted optical flow usually causes distortion of the result, so extra measures (Bao et al., 2019; 2018) are usually taken to refine the warped frame. Some methods generate intermediate frames directly, like CAIN (Choi et al., 2020) using channel attention and RIFE (Huang et al., 2022) using distillation to get a good result from one simple architecture. IFRNet (Kong et al., 2022) proposes a novel single encoder-decoder-based method to jointly perform intermediate flow estimation and intermediate feature refinement for efficient VFI, which also performs real-time inference with excellent accuracy. All video frame interpolation methods mentioned above cannot generate well on omnidirectional frames because they are not designed for them. Additionally, to our best knowledge, few methods have been proposed for Omni-VFI.

### 2.3 Conditional Modeling and Integration

Many real-world problems require the integration of multiple sources of information. When dealing with such issues, it often makes sense to process one source of information within the context provided by other prior knowledge. In the realm of machine learning, this contextual processing is often termed conditioning: the computations performed by a model are influenced or adjusted by prior knowledge derived from an auxiliary input. In Dumoulin et al. (2018), feature-wise transformations within many neural network architectures prove to be highly efficient in addressing a remarkably vast and diverse array of problems. Their success can be attributed to their adaptability in acquiring an effective representation of the conditioning input across various scenarios.

In the OSRT (Yu et al., 2023), a distortion-aware transformer is introduced for omnidirectional image super-resolution. This approach incorporates ERP distortion priors into the attention mechanism to adjust for distortion. Specifically, OSRT applies deformations to the feature maps by using offsets derived from latitude-dependent conditions, allowing continuous and adaptive modulation. This process enables the network to model and correct ERP distortion effectively. The Spatial Feature Transform (SFT) (Wang et al., 2018) leverages semantic segmentation probability maps as categorical priors to provide valuable conditioning information for image super-resolution. First, the low-resolution input is processed through a semantic segmentation network to generate probability maps, indicating the likelihood of each pixel belonging to different semantic categories (e.g., sky, buildings, vegetation, etc.). Then, the SFT layer uses these probability maps to learn scaling parameters specific to each category, applying them to modulate the intermediate feature maps. This approach enhances the realism and textural detail of the resulting high-resolution image by aligning it with its semantic context.

### 2.4 Omnidirectional Datasets

While datasets for plane VFI tasks have been extensively researched, including Vimeo90K (Xue et al., 2019), UCF101 (Soomro et al., 2012), and SNU-FILM (Choi et al., 2020), there is a lack of dedicated datasets for Omni-VFI. Leveraging existing datasets can streamline the benchmark creation process for Omni-VFI, ensuring standardization and reducing the learning curve for researchers. However, no dataset is used directly in the computer vision community for Omni-VFI. Thanks to the contributions of Wang et al. (2023) and S3PO (Baniya et al., 2023), high-resolution omnidirectional video super-resolution datasets ODV360, 360VDS, and 360UHD have been introduced, which can be utilized in Omni-VFI.

## 3 360VFI Dataset

To advance the development of Omni-VFI, a comprehensive dataset is paramount. We present a novel dataset curated from multiple sources and tailored specifically for Omni-VFI. Our dataset amalgamates three distinct collections, denoted as ODV360, 360VDS, and 360UHD, each contributing unique insights and challenges to the Omnidirectional Image Super-Resolution domain. We removed all dirty and unsuitable data and recollected it for the frame interpolation task.

### 3.1 ODV360, 360VDS and 360UHD

ODV360 (Wang et al., 2023) is a novel high-resolution (4K-8K) ODV dataset curated to address the scarcity of high-quality video datasets in the field of omnidirectional video super-resolution. It comprises 90 videos sourced from YouTube and public omnidirectional video repositories, alongside an additional 160 videos captured using Insta360 cameras, including models such as Insta 360 X2 and Insta 360 ONE RS. The dataset covers a diverse range of content spanning indoor and outdoor scenarios, downsampled to a standardized 2K resolution (2160x1080) for ease of use in ODV super-resolution.

360VDS and 360UHD are both proposed in S3PO (Baniya et al., 2023). They created a new dataset of Omnidirectional Videos specifically designed for super-resolution termed 360VDS. Open-source datasets used in other areas of 360° video research were assembled to create the 360VDS. Additionally, they also made use of the publicly available 360° video dataset from the Stanford VR lab called psych-360 (Miller et al.,

2020). They also created a 360 Ultra High-Definition (360UHD) dataset, consisting of eight clips ranging from HD to 4K.

### 3.2 Proposed 360VFI Dataset

Table 1: Comparisons Between Different Video datasets.

| | Modality | Task | Partition | Scenarios | | | |
|---|---|---|---|---|---|---|---|
| | | | | Indoor | Outdoor | People | Landscape |
| Vimeo90K (Xue et al., 2019) | Plane Video | VFI | ✗ | ✓ | ✓ | ✓ | ✗ |
| CAIN (Choi et al., 2020) | Plane Video | VFI | ✓ | ✓ | ✓ | ✓ | ✗ |
| ODV360 (Wang et al., 2023) | 360° Video | 360° VSR | ✗ | ✓ | ✓ | ✓ | ✓ |
| 360VDS (Baniya et al., 2023) | 360° Video | 360° VSR | ✗ | ✗ | ✓ | ✓ | ✓ |
| 360UHD (Baniya et al., 2023) | 360° Video | 360° VSR | ✗ | ✗ | ✓ | ✓ | ✓ |
| SUN360 (Torralba, 2012) | 360° Image | 360° OLSE | ✗ | ✓ | ✓ | ✓ | ✓ |
| 360-SOD (Zhao et al., 2023) | 360° Image | 360° OD | ✗ | ✗ | ✓ | ✗ | ✗ |
| **360VFI(Ours)** | 360° Video | 360° VFI | ✓ | ✓ | ✓ | ✓ | ✓ |

Table 2: Motion (latitude flow magnitude) statistics for each setting in 360VFI.

| | Easy | Middle | Hard | Extreme | All |
|---|---|---|---|---|---|
| Triplets | 518 | 260 | 76 | 76 | 930 |
| Extent | [0, 2] | [2, 3] | [3, 4] | [4, 10] | [0, 10] |

Our 360VFI dataset adopts a similar format as Vimeo90K (Xue et al., 2019). Each sample in our dataset consists of a triplet of video frames. Notably, the first and third frames are designated as input frames for the Omni-VFI model, while the second serves as the ground truth target frame. In the ODV360 (Wang et al., 2023) dataset, each video comprises 100 frames, while in the 360VDS and 360UHD (Baniya et al., 2023) dataset, the videos consist of 20 frames. We omit the final frame from each video in the ODV360 dataset, resulting in 99 frames per video, subsequently divided into 33 triplets. For videos in the 360VDS and 360UHD datasets, the first and last frames are discarded, leaving 18 frames per video, which are divided into six triplets. Each triplet constitutes an independent sample and the whole dataset consists of 930 triplets. This comprehensive arrangement enables a holistic evaluation of interpolation algorithms against genuine ODV content. We randomly divide these videos into training and testing sets.

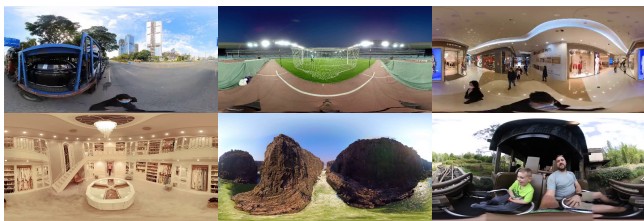

Figure 2: Examples of Different Scenarios in 360VFI Dataset

Videos in our dataset cover various scenarios, including natural landscapes, playgrounds, interiors of houses and cars, and indoor markets, as shown in Figure 2. To facilitate nuanced evaluations and benchmarking, we stratified the dataset into four distinct settings based on the varying degrees of motion inherent within the omnidirectional scenes, ranging from 0.19 to 9.69. These settings are categorized as easy, middle, hard, and extreme; the triplet numbers and their respective storage are illustrated in Table 2. The motion ranges for these settings are as follows: [0.19, 2], [2, 3], [3, 4], and [4, 9.69]. The frames and optical flow of different partitions are illustrated in Figure 3. This stratification allows researchers to systematically

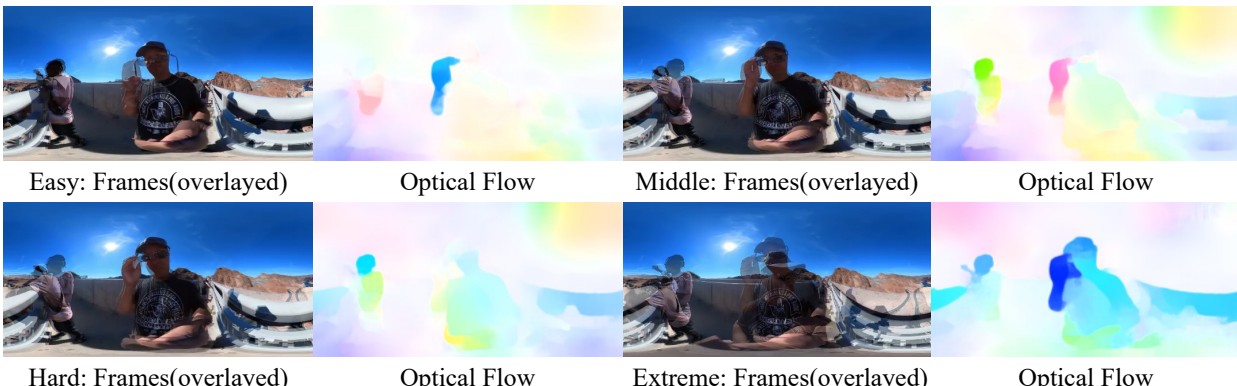

| Easy: Frames(overlayed) | Optical Flow | Middle: Frames(overlayed) | Optical Flow |
| Hard: Frames(overlayed) | Optical Flow | Extreme: Frames(overlayed) | Optical Flow |

Figure 3: Input Frames and Optical Flow of Different Settings in 360VFI Dataset. The colored parts in the optical flow image are larger and deeper when the motion is larger. The motion increases from the easy setting to the extreme setting.

assess the performance of their models across varying levels of motion complexity, facilitating a deeper understanding of model robustness and generalization capabilities. In Table 1[1], we compare the 360VFI dataset with other datasets. These datasets include plane video, 360° video, and 360° images. Our dataset is the first one composed of 360° video for omnidirectional video interpolation. Our dataset establishes a solid foundation for future research in Omni-VFI. By amalgamating diverse datasets and incorporating motion-based stratification, it not only enriches the existing repository of ODV data but also sets a precedent for structured evaluation and benchmarking in this evolving field. We anticipate that our dataset will serve as a catalyst for innovation and foster the development of more sophisticated and resilient omnidirectional video interpolation models in the future.

## 4 Method

Given two adjacent omnidirectional video frames $I_0$ and $I_1$, Omni-VFI aims at generating a non-existent intermediate omnidirectional frame $I_p$ to enhance visual quality and consistency between adjacent frames, which is as similar as possible to ground truth frame $I_g$. Current plane VFI methods use networks to directly learn a mapping function $G_\theta$ parametrized by $\theta$ as

$$I_p = G_\theta(I_0, I_1). \tag{1}$$

In order to generate $I_p$, a specific loss function $\mathcal{L}$ is designed for Omni-VFi to optimize $G_\theta$ on the training samples,

$$\hat{\theta} = \underset{\theta}{\arg\min} \sum_i \mathcal{L}(I_p, I_g), \tag{2}$$

where $(I_0, I_1, I_g)$ are training pairs.

We show that ERP distortion prior, i.e., knowing that the distortion in omnidirectional frames is latitude-dependent, is beneficial to generate a more accurate intermediate frame. The prior knowledge $\Psi$ can be conveniently represented by the ERP distortion condition map $C_d$ and we will explore more details about $C_d$ in Section 4.1,

$$\Psi = C_d. \tag{3}$$

To introduce priors in Omni-VFI, we reformulate equation 1 as

$$I_p = G_\theta(I_0, I_1 | \Psi), \tag{4}$$

---

[1]VFI: Video Frame Interpolation; VSR: Video Super-Resolution; OLSE: Outdoor Light Source Estimation; OD: Object Detection

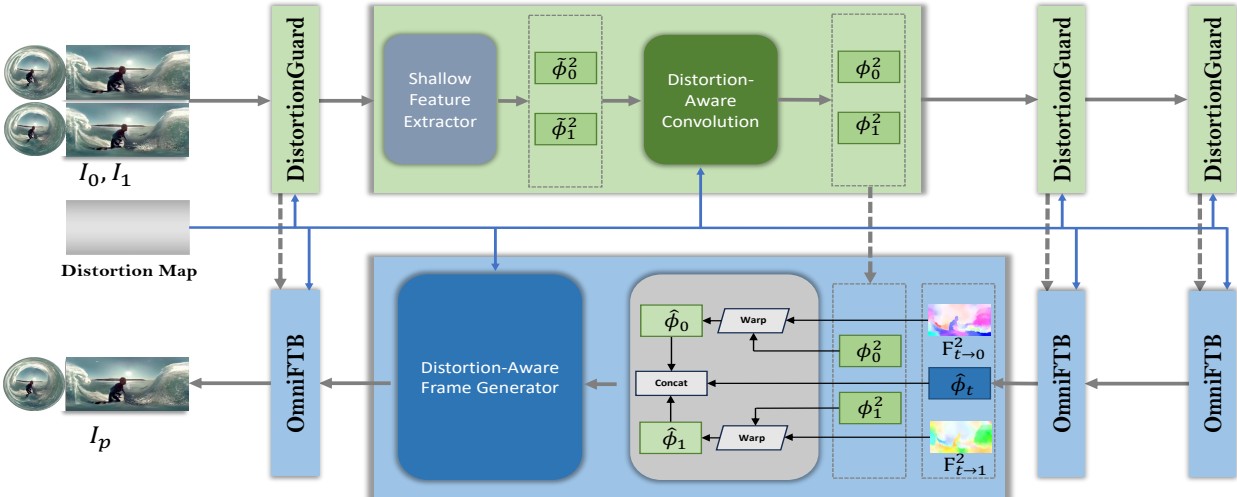

Figure 4: Architectural Overview. Our model is an efficient encoder-decoder based network, which first extracts less distorted pyramid context features $\phi_0^l$ and $\phi_1^l$ from input omnidirectional frames $I_0$, $I_1$ with DistortionGuards, and then gradually refines bilateral intermediate flow fields $F_{t \to 0}^l$ through OmniFTB generator, until yielding the target frame $I_p$. The figure above gives an illustration of the second-level DistortionGuard and OmniFTB, and details are shown below in Figure 6 and Figure 7.

where $\Psi$ defines the prior upon which the mapping function can condition. As mentioned in Section 2.3, there are many works using prior knowledge to guide their task in other computer vision tasks. Hence, we propose 360VFI Net, including DistortionGuard and OmniFTB.

## 4.1 Distortions in Omnidirectional Video

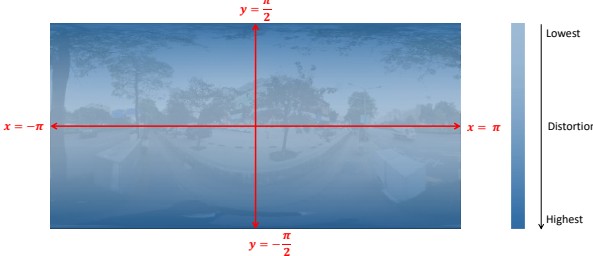

Figure 5: Illustration of an ERP frame and the distortion condition. The extent of distortion is the most severe in the polar regions.

ERP (Equirectangular Projection) is a commonly used method for mapping spherical surfaces onto a plane. In ERP, every point on the sphere is projected in such a way that each line from the polar center maintains the same distance. As a result, the distortion in ERP is latitude-dependent, with all pixels at the same height experiencing the same level of distortion. This makes ERP a practical choice for storing or transmitting omnidirectional images and videos, as it preserves depth and distance perception while minimizing distortions. However, the degree of distortion varies depending on the projection type, and ERP introduces significant distortions, especially at higher latitudes.

To quantify the extent of distortion in ERP, we follow the definition of the stretching ratio ($\mathbf{K}$) as described in Sun et al. (2017a), which measures the distortion at various points in relation to the ideal spherical surface. The stretching ratio $\mathbf{K}$ is determined by the variation in area between the spherical surface and the

projection plane. In ERP, the coordinates are defined as $x$ and $y$, and the stretching ratio can be expressed as:

$$\mathbf{K}_{\text{ERP}}(x, y) = \frac{\delta S}{\delta P} = \cos(y), \tag{5}$$

where $\delta S$ represents the area on the spherical surface and $\delta P$ represents the area on the projection plane, with $x \in (-\pi, \pi)$ and $y \in (-\frac{\pi}{2}, \frac{\pi}{2})$. From this equation, it becomes clear that ERP distortion depends solely on latitude, with the most severe distortion occurring in the polar regions.

For an input frame $X \in \mathbb{R}^{C \times M \times N}$, we can derive the distortion condition map $C_d \in \mathbb{R}^{1 \times M \times N}$ as follows:

$$\boldsymbol{C_d} = \cos\left(\frac{m + 0.5 - M/2}{M}\pi\right), \tag{6}$$

where $m$ is the height index of the input frame. This map helps capture the latitude-dependent distortion across the frame.

Spatial distortion is a major challenge in omnidirectional video interpolation due to the projection methods used, such as ERP. Objects near the image borders or poles suffer from significant stretching or compression, leading to inconsistencies in their size and shape across the video sequence. This latitude-dependent deformation complicates both the motion estimation and frame interpolation processes, making it difficult to maintain visual coherence. Temporal motion patterns also present a challenge in omnidirectional video interpolation. Accurately estimating object trajectories, velocities, and accelerations within the video sequence is complicated by the distortion present in ERP images, particularly as objects move from the center (low latitude) to the poles (high latitude). This distortion, referred to as depth distortion or spherical distortion, alters the appearance and size of objects, further complicating interpolation efforts.

## 4.2 Distortion Priors Using Approaches

### 4.2.1 DistortionGuard: A Distortion-Aware Pyramid Feature Extractor

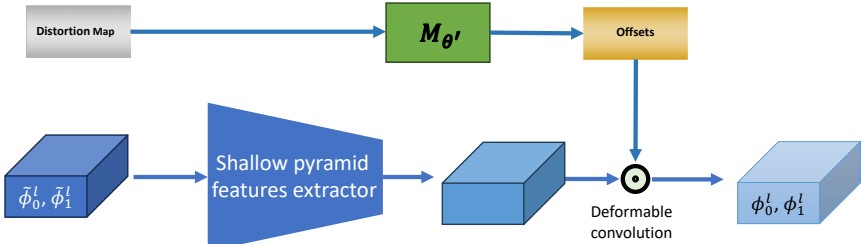

Figure 6: DistortionGuard (A Distortion-aware Features Extractor): The inputs are ERP frames or features $\tilde{\phi}_0^l$ and $\tilde{\phi}_1^l$ that are extracted from the last-level encoder. Then the extractor outputs less distorted features $\phi_0^l$ and $\phi_1^l$.

Previous methods tend to treat $C_d$ as an additional input of $X$ (Nishiyama et al., 2021), or resampling the convolution kernel at each location based on $C_d$ (Khasanova & Frossard, 2019). However, continuous and amorphous distortions cannot be adequately fitted by scattering and structured convolution operations. Therefore, we intend to design a novel block to learn distorted patterns continuously. In video frame interpolation tasks, the deformable mechanism is proposed to align features between adjacent frames (Tian et al., 2020; Wang et al., 2020b). Unlike standard DCN (Dai et al., 2017), which calculates offsets from the input feature map, offsets are calculated from bi-directional optical flow in VSR pipelines.

Drawing inspiration from feature-level flow warping techniques in Video Super-Resolution (VSR), Yu et al. (2023) employs feature-level warping operations to modulate Equirectangular Projection (ERP) distortion.

To learn distortions and extract features meanwhile, we propose the block DistortionGuard to modulate ERP distortion, as shown in $C_d$ is utilized to calculate the deformable offsets. More specifically, we apply a standard deformable convolution layer with a substituted input for offset calculation. Modulated less distorted features $\tilde{\phi}_0^l$ and $\tilde{\phi}_1^l$ are extracted as:

$$\tilde{\phi}_0^l, \tilde{\phi}_1^l = H_{\text{DCN}}(\tilde{\phi}_0, \tilde{\phi}_1, H_{\text{offset}}(C_d)), \tag{7}$$

where $H_{\text{DCN}(\cdot)}$ denotes standard deformable convolution layer (Zhu et al., 2019). DistortionGuard is designed to address the challenges posed by ERP distortion by incorporating prior knowledge of distortion patterns into feature extraction. By employing deformable convolution layers, DistortionGuard effectively modulates ERP distortion to enhance feature extraction.

### 4.2.2 OmniFTB: Omnidirectional Distortion Feature Transform Block

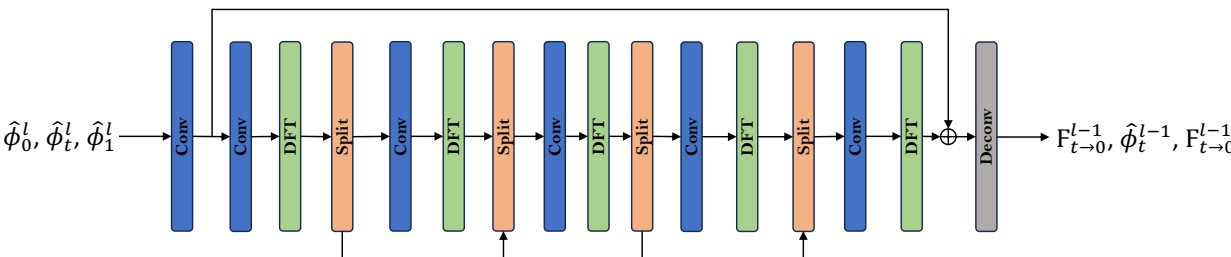

(a) Details of the Distortion-aware Frame Generator in Frame Generation Stage

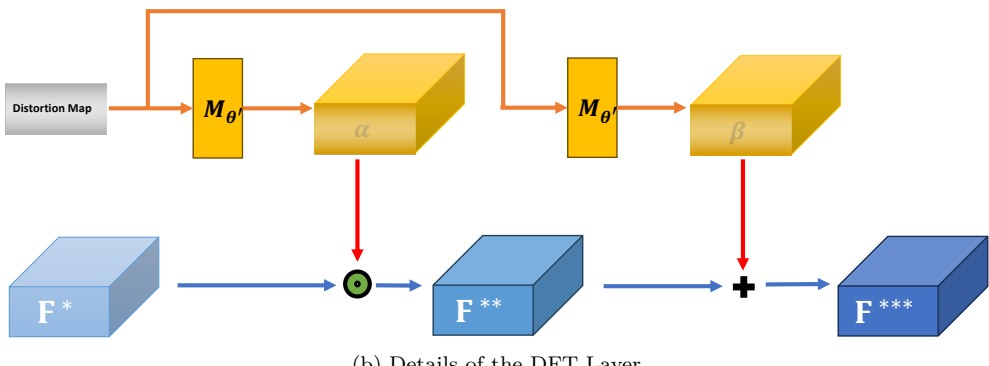

(b) Details of the DFT Layer

Figure 7: OmniFTB is a Distortion-aware Frame Generator: (a) shows the details of OmniFTB that are composed of Distortion-aware Feature Transform (DFT) layers and other parts; (b) shows the details of a DFT layer using affine transformation. It transforms the feature $\boldsymbol{F}^*$ from the last convolution layer with the parameter from the ERP distortion map into the feature with distortion $\boldsymbol{F}^{***}$. Hence, DFT layers recover ERP distortion from less distorted features.

Unlike DistortionGuard, which uses priors in the feature extraction stage, the affine transformation efficiently performs feature-wise transformations (Dumoulin et al., 2018). Consequently, in the intermediate feature reconstruction of our method, we propose another block considering ERP priors.

The distortion is modulated successfully in the output feature from the extraction stage. So, we devise a distortion-aware feature transform block (OmniFTB) to apply an affine transformation with a pair of distortion-based parameters. In OmniFTB, a mapping function $M_{\theta'}$ based on ERP distortion condition map $\boldsymbol{C_d}$ is learned at first,

$$\boldsymbol{M_{\theta'}} : \boldsymbol{C_d} \mapsto (\boldsymbol{\alpha}, \boldsymbol{\beta}), \tag{8}$$

where $(\boldsymbol{\alpha}, \boldsymbol{\beta})$ is the parameters to apply affine transformation. After obtaining $(\boldsymbol{\alpha}, \boldsymbol{\beta})$ from conditions, the transformation is carried out by scaling and shifting feature maps in the proposed DFT block:

$$\boldsymbol{F}^{***} = \boldsymbol{DFT}(\boldsymbol{F}^*|\boldsymbol{C_d}) = \boldsymbol{DFT}(\boldsymbol{F}^*|\boldsymbol{\alpha}, \boldsymbol{\beta}) = \boldsymbol{\alpha} \odot \boldsymbol{F}^* + \boldsymbol{\beta}, \tag{9}$$

where $\boldsymbol{F}^{***}$ denotes the intermediate feature maps, whose dimension is the same as $\boldsymbol{\alpha}$ and $\boldsymbol{\beta}$, and $\odot$ is referred to Hadamard product, i.e., element-wise multiplication. The DFT layer simultaneously performs feature-wise manipulation but also spatial-wise transformation since the spatial dimensions are preserved. Figure 7. (b)shows an example of implementing DFT block embedded in the intermediate frame generating decoder. The mapping function $M_{\theta'}$ can be arbitrary learnable functions. In this study, we use a neural network for $M_{\theta'}$ to optimize it end-to-end with the Omni-VFI branch. OmniFTB complements DistortionGuard by focusing on intermediate feature reconstruction with distortion. By applying an affine transformation with distortion-based parameters, OmniFTB effectively leverages ERP distortion priors to transform feature maps for accurate intermediate frame generation.

## 5 Experiments

### 5.1 Implementation Details

We implement the proposed algorithm in PyTorch and utilize the 360VFI dataset to train the 360VFI network from scratch. Our model is optimized by AdamW (Loshchilov & Hutter, 2019) algorithm for 300 epochs with a total batch size of 8 on Two NVIDIA Tesla V100 GPUs. Initially, the learning rate is set to $1 \times 10^{-4}$ and gradually decays to $1 \times 10^{-5}$ following a cosine attenuation schedule. Throughout the training process, we refrained from applying augmentation techniques such as rotating and random cropping to triplet samples. This decision was made due to the rigid nature of latitude-dependent distortion.

### 5.2 Evaluation and Comparison

We conducted comparative experiments to evaluate the performance of 360VFI Net against existing methods. We utilized the first omnidirectional video dataset 360VFI we proposed and compared our method with several common VFI approaches. We adopt the WSS-L1 Loss (Baniya et al., 2023) as the loss function for training 360VFI Net. The Smooth L1 loss combines the strengths of both L1 and L2 losses, governed by a hyper-parameter $\beta$. It offers the benefits of L1 loss (steady gradients for large errors) and L2 loss (less oscillation for small errors), making it more robust to outliers and preventing exploding gradients in certain scenarios. However, traditional loss functions do not account for the unique characteristics of ERP frames, particularly the distortion that occurs across latitudes. This distortion can lead to significant prediction errors, especially in polar regions, when using learning-based models.

To address this issue, the Weighted Spherically Smooth-L1 (WSS-L1) Loss (Baniya et al., 2023) was introduced, building on the idea of WS-PSNR. The WSS-L1 Loss adjusts the Smooth L1 loss to account for ERP distortion, as shown in the following equation:

$$\textbf{WSS-L1}\boldsymbol{Loss} = \begin{cases} \left(\frac{0.5(GT-HR)^2}{\beta}\right) \times \psi, & \text{if } |GT - HR| < \beta \\ (|GT - HR| - 0.5\beta) \times \psi, & \text{otherwise} \end{cases} \tag{10}$$

Here, $\psi$ represents the ERP distortion map, and $\beta$ is the hyper-parameter that controls the transition between L1 and L2 behavior.

We compared the proposed 360VFI Net module with the following methods: IFRNet (Kong et al., 2022), DQB (Zhou et al., 2023), EMA-VFI (Zhang et al., 2023), EBME (Jin et al., 2023b) and UPR-Net (Jin et al., 2023a). We employed the following evaluation metrics to assess the performance of different interpolation methods: Peak Signal-to-Noise Ratio (PSNR), Structural Similarity Index (SSIM), Weighted PSNR (WS-PSNR) (Sun et al., 2017b) and Weighted Structural Similarity Index (WS-SSIM) (Zhou et al., 2018). WS-PSNR extends PSNR by considering the importance of different regions in ODV and calculating PSNR

values with weighting factors to emphasize regions of interest, such as central regions with high visual significance. Similarly, WS-SSIM extends SSIM by incorporating weighted factors to account for the perceptual importance of different regions in ODV. It provides a more accurate assessment of perceptual quality considering distortion. With the comprehensive evaluation metrics, we aim to thoroughly evaluate the performance of Omni-VFI methods on four settings of different motion extents, considering both fidelity to ground truth and perceptual quality across the entire omnidirectional view.

Table 3: Quantitative comparison of Omni-VFI results on the dataset 360VFI

| Method | Easy | | Middle | | Hard | | Extreme | |
|---|---|---|---|---|---|---|---|---|
| | PSNR↑ SSIM↑ | WS-PSNR↑ WS-SSIM↑ | PSNR↑ SSIM↑ | WS-PSNR↑ WS-SSIM↑ | PSNR↑ SSIM↑ | WS-PSNR↑ WS-SSIM↑ | PSNR↑ SSIM↑ | WS-PSNR↑ WS-SSIM↑ |
| **IFRNet** Kong et al. (2022) | 34.38/0.9685 | 33.90/0.9538 | 28.03/0.9136 | 28.89/0.9047 | 26.91/0.8905 | 27.74/0.8724 | 24.43/ 0.8421 | 24.80/0.7999 |
| **DQBC** Zhou et al. (2023) | 34.06/0.9622 | 33.51/0.9449 | 26.33/0.8891 | 27.35/0.8776 | 25.14/0.8567 | 26.02/0.8329 | 22.81/0.7967 | 23.26/0.7410 |
| **EMA-VFI** Zhang et al. (2023) | 33.92/0.9681 | 33.40/0.9529 | 27.45/0.9112 | 28.37/0.9024 | 26.88/0.8985 | 27.64/0.8817 | 24.26/0.8493 | 24.89/0.8161 |
| **EBME** Jin et al. (2023b) | 33.93/0.9613 | 33.48/0.9447 | 26.27/0.8884 | 27.35/0.8772 | 25.08/0.8561 | 25.97/0.8318 | 22.84/0.7975 | 23.23/0.7403 |
| **UPR-Net** Jin et al. (2023a) | 34.03/0.9686 | 33.45/0.9535 | 27.47/0.9116 | 28.42/0.9024 | 26.89/0.8988 | 27.72/0.8816 | 24.34/0.8507 | 24.85/0.8157 |
| Ours | 34.48/0.9687 | 33.95/0.9537 | 28.13/0.9154 | 28.96/0.9060 | 27.41/0.9028 | 27.81/0.8879 | 25.52/0.9021 | 25.63/0.8517 |

### 5.2.1 Quantitative comparison

The evaluation results, summarized in Table 3, demonstrate the effectiveness of our 360VFI Net compared to competing methods. Our method consistently achieves superior performance across all metrics evaluated, especially in hard and extreme settings, showcasing its ability to generate high-quality omnidirectional frames in large vertical motion. Specifically, the higher PSNR and SSIM scores indicate better reconstruction fidelity and perceptual quality of the interpolated frames. Moreover, the incorporation of weighted metrics, WS-PSNR and WS-SSIM, further highlights the robustness of our method in handling distortions and variations in ODV sequences.

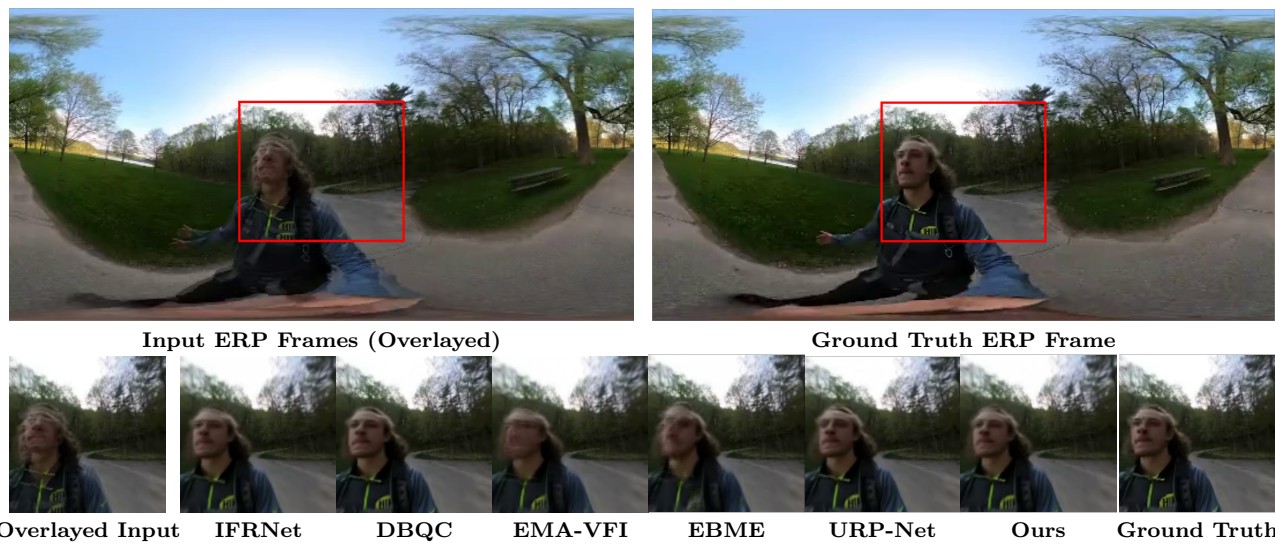

**Input ERP Frames (Overlayed)**        **Ground Truth ERP Frame**

**Overlayed Input** **IFRNet** **DBQC** **EMA-VFI** **EBME** **URP-Net** **Ours** **Ground Truth**

Figure 8: Qualitative comparisons visualization of five previous SOTA VFI approaches

### 5.2.2 Qualitative comparison

In addition to quantitative evaluation, we conducted a qualitative comparison to visually assess the performance of our proposed 360VFI Net compared to other state-of-the-art methods in plane VFI. We present a figure below illustrating the generated intermediate frames by different methods alongside the ground truth intermediate frames. As depicted in Figure 8, our 360VFI Net consistently produces visually pleasing and high-quality intermediate frames that closely resemble the ground truth frames. The results interpolation exhibits smooth transitions and preserves details effectively, demonstrating the robustness and efficacy of our method in handling Omni-VFI tasks. In contrast, the results from other methods may suffer from artifacts or distortion, leading to less faithful representations of the ground truth frames.

### 5.3 Ablation Study

Table 4: Ablation Study on Proposed Block

| DistortionGuard | OmniFTB | Easy | | Middle | | Hard | | Extreme | |
|---|---|---|---|---|---|---|---|---|---|
| | | PSNR↑ SSIM↑ | WS-PSNR↑ WS-SSIM↑ | PSNR↑ SSIM↑ | WS-PSNR↑ WS-SSIM↑ | PSNR↑ SSIM↑ | WS-PSNR↑ WS-SSIM↑ | PSNR↑ SSIM↑ | WS-PSNR↑ WS-SSIM↑ |
| × | × | 33.65/0.9413 | 33.32/0.9401 | 27.51/0.9006 | 27.55/0.8997 | 26.57/0.8963 | 26.76/0.8792 | 24.63/0.8947 | 24.69/0.8401 |
| × | ✓ | 33.97/0.9659 | 33.64/0.9480 | 27.73/0.9072 | 28.14/0.9022 | 27.05/0.8965 | 27.29/0.8836 | 25.02/0.8981 | 25.09/0.8522 |
| ⊛ | ✓ | 34.00/0.9653 | 33.87/0.9509 | 27.81/0.9078 | 28.36/0.9046 | 27.12/0.8965 | 27.44/0.8840 | 25.06/0.8980 | 25.23/0.8550 |
| ✓ | × | 34.05/0.9565 | 33.66/0.9487 | 27.79/0.9087 | 28.24/0.9036 | 27.19/0.8990 | 27.27/0.8830 | 25.00/0.8975 | 25.14/0.8529 |
| ✓ | ✓ | 34.48/0.9687 | 33.95/0.9537 | 28.13/0.9154 | 28.96/0.9060 | 27.41/0.9028 | 27.81/0.8879 | 25.52/0.9021 | 25.63/0.8617 |

In order to test the effectiveness of the two key components, DistortionGuard and OmniFTB, within the 360VFI Net architecture, we performed a series of ablation experiments. In these experiments, we systematically removed each module and evaluated its impact on overall performance. Specifically, for the DistortionGuard ablation, we created a variant of 360VFI Net without the deformable convolution, one with a normal convolution ($\times$) and one resampling the convolution kernel at each location based on the static distortion parameters ($\circledast$). Then we compared its performance against the total 360VFI Net. For the OmniFTB ablation, we constructed a variant of the model without the DFT block and contrasted it with the complete architecture to demonstrate that affine transformation is aware of this distortion.

Both experiments were conducted under identical datasets and training conditions. As shown in Table 4, the removal of either module resulted in a significant performance degradation compared to the full model, underscoring the critical role that both DistortionGuard and OmniFTB play in achieving optimal results.

The ablation results clearly demonstrate the substantial contribution of the DistortionGuard and OmniFTB modules to the 360VFI network's overall performance. Their inclusion enhances the model's robustness and accuracy, playing an essential role in omnidirectional video frame interpolation tasks. These findings further validate the effectiveness and reliability of the 360VFI Net architecture, providing a solid foundation for future research and development in this domain.

## 6 Conclusion

360VFI is the first dataset and benchmark for omnidirectional video frame interpolation. Our dataset has four evaluation settings, serving as a benchmark for various extents of motion in omnidirectional videos and facilitating future research in this field. Furthermore, the proposed 360VFI network uniquely incorporates ERP distortion priors in both the feature extraction stage and the frame generation stage, employing different customized methods for each stage. It first extracts less distorted features of two input frames according to the distortion map and then gradually generates the target ERP frame using parameterized affine transformation to recover the distortion. We anticipate our contributions will inspire further advancements in omnidirectional video processing techniques, ultimately enhancing the immersive viewing experience.

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
