# OpenReview forum: "360VFI: A Dataset and Benchmark  for Omnidirectional Video Frame Interpolation"
_TMLR — Rejected by TMLR_

### Review · Reviewer_NbYo · 2024-10-10

**Summary Of Contributions:**

The paper introduces a new 360 video benchmark dataset, 360VFI. It has high diversity of content type and motion patterns and is suited for omnidirectional video frame interpolation and cv tasks in omnidirectional domains.  Additionally, a novel method for omnidirectional video interpolation, Omni-VFI is proposed in this paper, which the main novelty be the introduction of ERP distortion prior. The distortion prior is based on the fact that omnidirectional frames captured by 360 camera has latitude dependent distortion due to the camera setting and how 360 image is mapped to 2D image plane.

**Audience:**

Yes

**Broader Impact Concerns:**

There is no concerns on broader implications.

**Claims And Evidence:**

Yes

**Requested Changes:**

1. Currently all experiments are using the same 360VFI dataset, please add other benchmark datasets to show the proposed method are better across different settings.
2. Regards to ERP distortion prior, such distortion in mainly introduced when 360 image is projected to 2d image frames. One way to avoid such distortion is to project 360 image to multiple 2d images instead of one image. I would like to see some discussion about such approach compared to using a distortion prior to fix artificially introduced distortions.

**Strengths And Weaknesses:**

Strengths:

1. provides a new and diverse benchmark dataset for omnidirectional vision domains. This is a welcome move given the limited data in such settings.

2. The introduction of distortion prior for handle 360 images is a simple but effective method to use existing 2D CV methods in omnidirectional settings

Weakness:
1. Regards to ERP distortion prior, such distortion in mainly introduced when 360 image is projected to 2d image frames. One way to avoid such distortion is to project 360 image to multiple 2d images instead of one image. I would like to see some discussion about such approach compared to using a distortion prior to fix artificially introduced distortions.

---

> ### Author Response · Authors · 2024-12-30
> **Response to Reviewer NbYo**
>
> Thank you for your review. The requested changes are addressed as follows.
>
> 1. We would like to clarify that the 360VFI dataset is the first video frame interpolation dataset for omnidirectional videos. Our dataset is comprehensive, including 930 triplets in four settings. We would like to add other benchmark datasets if new benchmarks are available.
>
> 2. There are generally 3 common methods to project 360 images to 2D images, equirectangular Projection, tri-cylindrical Projection and cube-padding Projection. Tri-cylindrical approach projects one 360 image to three 2D images. However, both tri-cylindrical Projection and cube-padding Projection are not used in the common dataset. The reasons are twofold. First, omnidirectional images and videos are inherently spherical, resembling a globe. To store and transmit these data efficiently, we need to project them into 2D images. ERP is the most suitable format to review by humans and easily back-project to 360 format [1] and both tri-cylindrical Projection and cube-padding Projection need more computational cost. Second, omnidirectional images or videos that can be found on the Internet are almost all in ERP, so it is natural to develop 360 video processing techniques only in the ERP format. Although Li et al. [2]  explore 360 optical flow estimation based on considering different projections, they only need ERP in the inference stage instead of the training stage. Consequently, it is impractical to train a frame interpolation model in other projections because of data and efficiency.
> Thank you again for your valuable feedback.
>
> [1] The equation of projection and back-projection can be found in the paper;
>
> [2] https://arxiv.org/abs/2208.00776

---

### Review · Reviewer_MTuW · 2024-11-20

**Summary Of Contributions:**

The contributions of this paper can be summarized in two parts:
- Proposal of a new dataset: This introduces a new dataset for frame interpolation in omnidirectional videos. The proposed dataset, named 360VFI, encompasses a wide range of content and motion patterns, providing a comprehensive benchmark for evaluating interpolation methods in the omnidirectional domain.
- Development of a simple VFI method: This also proposes a straightforward VFI method trained on the 360VFI dataset. This method outperforms existing plane video frame interpolation techniques by incorporating omnidirectional priors during both feature extraction and frame generation. This is particularly advantageous for scenarios involving large motion in omnidirectional frame interpolation.

**Audience:**

Yes

**Broader Impact Concerns:**

Not applicable.

**Claims And Evidence:**

Yes

**Requested Changes:**

1. In related work, the authors noted that few methods have been proposed for Omni-VFI, meaning that frame interpolation for Omni-VFI has not been as extensively studied as it has for typical videos. However, there are still some approaches. Could you share some of the methods for frame interpolation in Omni-VFI?
2. How to remove dirty or irrelevant video clips when constructing this dataset? Additional information on data cleansing would be helpful for fellow researchers.
3. I_1 and I_2 should be changed to I_0 and I_1 to match the labeling in figure 1. Additionally, what does the superscript 2 in \tilde{\phi}_1^2 represent? It doesn’t appear to indicate squaring.

**Strengths And Weaknesses:**

**Strengths**

Introducing a dataset for omnidirectional video tasks is always an appealing contribution. Moreover, the proposed method is specifically designed to address the absence of reliable temporal correspondences in omnidirectional video processing, significantly enhancing its applicability and robustness. While the blending of conditions, as seen in the FiLM approach, is not a novel concept, it proves to be effective in this particular task.

**Weaknesses**

My primary concern with this paper lies in the strengths of the proposed dataset. According to the paper, existing omnidirectional video datasets are unsuitable for frame interpolation tasks because many have low frame rates, which can lead to visual fatigue. However, based on my understanding, the 360VFI dataset has higher frame rates and greater diversity. This raises concerns about the limited advantages of 360VFI compared to the ODV360 dataset.

It appears that ODV360 can still be used to train frame interpolation models, as each video in the dataset consists of 100 frames. If this is the case, does the 360VFI dataset truly offer greater diversity than ODV360? According to Table 1, the coverage of scenarios (indoor, outdoor, people, landscape) is the same for both datasets.

Therefore, it is necessary to explicitly describe the unique strengths of the 360VFI dataset in the revised manuscript to clarify its advantages over existing datasets.

---

> ### Author Response · Authors · 2024-12-30
> **Response to Reviewer MTuW**
>
> Thank you for your review. We have addressed the requested changes as follows:
>
> 1. We apologize for the insufficient survey. However, to the best knowledge, we did not find any paper working on Omni-VFI. Could you please provide more information? If there are other Omni-VFI methods, we will be delighted to include a discussion in the paper.
> 2. All the omnidirectional video frames in our dataset are in the ERP format, which inherently exhibits latitude-dependent distortion. However, some frames contain a watermark in the form of a strip concatenated to the bottom of the frame, and some are rotated 90 degrees. This hurts the latitude-dependent distortion property, so we manually removed these frames. This process is not overly time-consuming, as we only need to inspect the first frame of each video rather than every triplet.
> 3. Thank you for pointing this out. We will make the necessary corrections in the updated paper. l in \tilde{\phi}_1^l means the l level encoder. So superscript 2 in Figure 4 represents the second encoder/decoder level.

---

### Review · Reviewer_fabx · 2024-12-17

**Summary Of Contributions:**

This paper addresses the problem of 360-degree video frame interpolation. To this end, the authors constructed a 360 video dataset that features diverse scenes and motion magnitudes. To deal with the spatially varying distortion of the equirectangular map, the authors also proposed a distortion-aware encoder-decoder architecture. The offset of the deformable convolution kernel is generated from a distortion map in the encoder (DistortionGuard), while an affine transformation is generated from the distortion map in the decoder (OmniFTB). To evaluate the effectiveness of the proposed method and dataset, the authors conducted both qualitative and quantitative comparisons and demonstrated superior performance over the baselines designed for regular perspective videos with narrower FOV.

**Audience:**

Yes

**Broader Impact Concerns:**

This paper does not need to have the Broader Impact Statement since the video frame interpolation problem does not change or generate new content of the video.

**Claims And Evidence:**

Yes

**Requested Changes:**

- First, please refer to the weaknesses section for the requested changes.
- Add explanation about the "split" operation in Figure.7 (a) which is missing in the paper.

**Strengths And Weaknesses:**

Strengths:
- This paper explores the novel problem of 360-degree video frame interpolation, and the corresponding solutions proposed could potentially be generalized to other 360-degree image/video applications.

- The paper is well-organized, with clear explanations of the motivation and its relation to existing works.

- The methods are introduced comprehensively, with observations and insights behind the design.

- The overall performance is superior to baselines, both quantitatively and qualitatively, verifying the effectiveness of the proposed method.

Weaknesses:
- The major weakness of the paper is the lack of justification for the design of the proposed architectures:
  - The learned offsets of the deformable convolution kernel should be further verified. Firstly, it is important to differentiate the learned offsets from fixed offsets that resample/interpolate convolution kernel based on the mapping from the equirectangular map to the tangent plane. For example, the authors should measure how the learned offsets differ from resampling a kernel to perform convolution on a tangent plane and also compare their performances in ablation studies. Secondly, it is also critical to visualize the learned offsets and provide intuitive and qualitative explanations about the mechanism of the learned offsets. Lastly, the offsets are fixed once learned since they only depend on the distortion map; therefore, it would be beneficial to discuss concatenating image features as well for offset prediction, which is closer to the original version of the Deformable Convolutional Network (DCN).
  - The affine transformation in Section 4.2.2 also needs further verification. Specifically, the operation in Equation 9 is an affine transformation with only channel-wise scale and shift; therefore, the operation is conducted at each single pixel at each pyramid level. It is unclear how this pixel-wise operation can recover the distortion that is corrected during the encoder stage to output the frame interpolation.
- Another weakness is the justification of the proposed dataset. The authors conducted experiments only on the proposed dataset to prove the effectiveness of the proposed method. However, the benefits of the dataset itself are not presented. Therefore, the authors should also train their proposed model with existing 360-degree video datasets to demonstrate the contribution of the dataset itself.

---

> ### Author Response · Authors · 2024-12-30
> **Response to Reviewer fabx**
>
> Thank you for your review. The requested changes are addressed as follows.
>
> 1. Please let us present the explanation of the proposed architectures:
>
> * First, resampling a kernel to perform convolution on a tangent plane is extremely computationally impractical especially for videos, as it requires complex projection at each frame, significantly increasing the processing time. Second, for a better understanding of the learned offset mechanism, please refer to Fig. 4 (b) of [1], which provides a detailed visualization of this process. Last, the offsets are fixed once learned but they depend on both the distortion map and the input domains (two frames). The VFI model learns a mapping function from two adjacent frames to one intermediate frame. 360VFI learns to map two omnidirectional frames to one intermediate omnidirectional frame. The input and output domains of VFI and Omni-VFI are different. The learned offsets are just a constraint to the learned model to perform better. However, we cannot obtain the real intermediate frame. The learned offsets help the model map the output of the encoder to a feature without distortion.
>
> * We are inspired to use the affine transformation to recover the distortion because it has been verified effective in other low-level vision problems [2]. Therefore, we tend to utilize this pixel-wise operation for recovering the distortion in the generated intermediate frames. Indeed, it is unclear how this pixel-wise operation can recover the distortion, however, we can find it effective in the ablation studies.
>
> 2. We would like to clarify that the 360VFI dataset is the first omnidirectional video frame interpolation dataset, which includes a comprehensive collection of all publicly available omnidirectional videos on the internet. Our dataset has four settings, covering abundant videos in various kinds of scenarios and different latitude motion extent, which is suitable to test the effectiveness of the Omni-VFI model. To the best of our knowledge, there does not exist a 360-degree video dataset for frame interpolation.
>
> 3. The "split" operation in Fig. 7 (a) refers to tensor slicing within the residual block, positioned between the first convolution and the final deconvolution, to adjust the channel dimensions of the tensors.
>
> Thanks again for your valuable review.
>
> [1] https://arxiv.org/abs/2302.03453
>
> [2] https://arxiv.org/abs/1804.02815

---

> > ### Comment · Reviewer_fabx · 2025-01-10
> >
> > Dear authors, thank you for the rely. I would like to further clarify my concerns as follows:
> > - The question I asked is not about resampling kernel for "every frame", instead it is about resampling the convolution kernel at each location based on the static distortion parameters, or you can do the cube maps to convert the equirectangular map to six perspective images, or you can do undistort small image patches into perspective image patches along it tangent plane. All these  solutions are similar, the only difference is about whether the "static and constant" undistortion happened on convolutional kernel or on the image. The question is about whether this "static and constant" undistortion is necessary to be learnt or only leverage the mapping between equirectangular mapping and perspective mapping (on each patches).
> > - For the question about affine mapping, [2] has a clear definition about the spatial transformation in the paper and also demonstrate that through ablation studies. This paper should do the same (define spatial transformation with distortion and also demonstrate that the affine transformation is aware of this distortion) since the spatial arrangement of equirectangular map is different.
> > - Any kind of 360 video data can be used as training for interpolation task, the matter is whether it is effective. The comparison will not diminish the contribution of this paper, instead this is to consolidate the contribution as the first dataset for 360 video interpolation.

---

> ### Author Response · Authors · 2025-01-15
>
> Dear Reviewer,
>
> Thanks for your detailed feedback and clarifications.
>
> 1. The "static and constant" undistortion itself does not need to be learned, as it can be directly derived from the mapping between equirectangular projection and perspective projection. However, our task focuses on video frame interpolation, where the process of generating intermediate frames inherently involves handling undistortion. In this context, the generation of intermediate distorted frames requires a learning-based approach to simultaneously model the motion and distortion effects. We will revise this discussion in Sec.4.2.2 and incorporate an ablation experiment in the updated paper to address this point.
> 2. The spatial transformation with distortion is defined in the paper and the ablation study demonstrates that the affine transformation is aware of this distortion; we will optimize the expression in Sec.4.2.2 and 5.3.
> 3. We have collected all the publicly available omnidirectional videos we could find and organized them into four settings based on the extent of vertical motion to demonstrate the effectiveness. While we aimed to establish a foundational dataset for 360-degree video interpolation, we agree that additional comparisons with existing datasets could further strengthen our contribution. If you could provide or suggest any existing 360-degree video datasets, we would be delighted to include comparisons.
>
> Thank you again for your insightful comments and suggestions.

---

### Review · Reviewer_x7Y6 · 2024-12-25

**Summary Of Contributions:**

This paper propose a dataset and benchmark for omnidirectional video frame interpolation, which clearly benefits the research community. It also propose a video frame interpolation method for omnidirectional videos. The core of the design lies on the omnidirectional distortion prior. The writing is good and the contributions are clear presented.

**Audience:**

Yes

**Broader Impact Concerns:**

There are no broader impact concerns in this paper.

**Claims And Evidence:**

Yes

**Requested Changes:**

Please refer to the weaknesses section.

In related work section 2.2, the author mentioned “few methods have been proposed for Omni-VFI.” If there are existing Omni-VFI methods, more explanation and comparison are necessary.

**Strengths And Weaknesses:**

Strengths
1. The first dataset and benchmark for omnidirectional video frame interpolation. I’m not familiar with this area. The fact needs to be checked.
2. Proposing a novel method designed for learning-based omnidirectional VFI. The design utilizes the inductive bias of the distortion prior.
3. The resolution of the collected 360 videos is high.

Weaknesses
1. The distortion for omnidirectional videos should be homography transformation instead of affine transformation. Why do the authors choose affine transformation?
2. The training and test sets are “randomly” divided. More details need to be supplemented.
3. The number of videos is too small, prohibiting its effectiveness.
4. The major contribution of this paper is the proposed dataset but the details about the dataset are absent. The authors need to present more statistics about the dataset, such as semantic categories, indoor and outdoor numbers, etc.

---

> ### Author Response · Authors · 2024-12-30
> **Response to Reviewer x7Y6**
>
> Thanks a lot for your helpful review.
>
> 1. Homography transformation, while more general and capable of capturing global perspective changes, might be overly complex and less efficient for the localized distortions present in omnidirectional videos. Additionally, prior low-level vision research, such as [1], has successfully utilized affine transformation for image super-resolution, which inspired our approach to distortion recovery.
>
> 2. The training and test sets are divided "randomly" through the following process: first, we shuffle the triplets using random.shuffle( ). Second, we split the triplets according to a predetermined proportion. Finally, the test triplets are moved to a separate file.
>
> 3. We would like to clarify that the 360VFI dataset comprises a comprehensive collection of all publicly available omnidirectional videos found on the internet.
>
> 4. The paper includes the necessary details about the 360VFI dataset. Regarding semantic categories are less relevant to Omni-VFI because omnidirectional videos are special in the motion pattern. For omnidirectional videos, it matters that the motion across different latitude parts of frames because the distortion is latitude dependent. Therefore, we structured the dataset based on different motion patterns.
>
> 5. Thank you for the feedback. To the best of our knowledge, we did not find any papers addressing Omni-VFI. We will be delighted to include a discussion in the updated paper if there are existing Omni-VFI methods,
>
> [1] https://arxiv.org/abs/1804.02815

---

> > ### Comment · Reviewer_x7Y6 · 2025-01-14
> >
> > Thank you for your reply.
> > 1. Homography is not much complex than affine, with only 1 more variable. Prior work utilizes affine transformation does not justify the reasonability of affine transformation in this paper. More important, it is expected to be tomography by definition.
> > 2. What is the predetermined proportion?
> > 3. However, the number of data does not fit modern computer vision datasets. How do you check ``all publicly available omnidirectional videos found on the internet''?

---

> > > ### Author Response · Authors · 2025-01-15
> > >
> > > Dear Reviewer,
> > >
> > > Thanks for your valuable suggestions. We sincerely appreciate your feedback, which helps us improve our work.
> > >
> > > 1. **Homography vs. Affine Transformation**
> > >    We acknowledge your comment regarding the complexity and applicability of Homography compared to Affine transformation. In this work, we initially chose Affine transformation for its simplicity and computational efficiency. We agree that the potential benefits of Homography require further investigation. In future work, we will explore the use of Homography, comparing its performance with Affine transformation in the context of our proposed method.
> > >
> > > 2. **Predetermined Proportion**
> > >    The predetermined proportion refers to the data split in our experiments: 80% of the data is used for training, and 20% is used for testing.
> > >
> > > 3. **Data Sources**
> > >    To collect omnidirectional videos, we searched several platforms and resources, including Google, Papers with Code, GitHub, ArXiv, and IEEE Xplore, using keywords such as "omnidirectional video datasets" and "360-degree video datasets." We welcome any recommendations for additional omnidirectional videos or datasets. If you could provide such resources, we would be delighted to include them in our experiments.
> > >
> > > We hope this response addresses your concerns. Thank you again for your constructive feedback.

---

### Decision · Action_Editor_UytC · 2025-02-03

**Recommendation:** Reject

**Comment:**

Please see above.

**Audience:**

Yes, some audience on video interploation and 360 video processing will be inerested.

**Claims And Evidence:**

This paper propose a dataset and benchmark for omnidirectional video frame interpolation, which will benefit the research community on video processing. It also proposes a video frame interpolation network for omnidirectional videos. The core of the design of the network lies on the omnidirectional distortion prior.

After the rebuttal, two reviewers are leaning to accept the paper while the other two are leaning to reject. The AE agrees with the concerns raised by the reviewers. The paper didn't meet the criterion of TMLR on providing convincing evidence:

- The major claim of this paper is the proposed dataset and benchmark for omnidirectional video, but it only includes 930 triplets and some triplets belong to the same video. The analysis and details of the proposed dataset are insufficient. It only classifies the videos into four difficulty levels according to the motion magnitude but more statistical information is expected. For instance, Vimeo-90K consists of 89,800 high-quality video clips, it provides histograms about pixel-wise flow frequency and Image mean flow frequency. Moreover, the videos are extracted from ODV360, 360VDS, 360UHD datasets, and are simply put together, which are already publicly available. Such a claim of a new dataset and benchmark being first of its kind is not fully justified.

**Resubmission Of Major Revision:**

The authors may consider submitting a major revision at a later time.